# Scalable Permutation-Aware Modeling for Temporal Set Prediction

## Abstract

Temporal set prediction involves forecasting the elements that will appear in the next set, given a sequence of prior sets, each containing a variable number of elements. Existing methods often rely on complex architectures with substantial computational overhead, limiting their scalability. In this work, we introduce a novel and scalable framework that combines an efficient input representation with permutation-equivariant and permutation-invariant transformations to model set dynamics. Our approach significantly reduces training and inference time while maintaining competitive performance. Extensive experiments on multiple public datasets demonstrate that our method achieves state-of-the-art performance overall, outperforming or matching existing models across several evaluation metrics. These results highlight the effectiveness of our model in enabling efficient and scalable temporal set prediction.

## 1   Introduction

Temporal Set Prediction addresses the problem of predicting which elements belong to the next set, given a sequence of sets. The problem involves identifying patterns in how sets evolve over time—tracking which elements enter, exit, or remain—and using these patterns to make accurate membership predictions. This approach enables fine-grained, element-level forecasting in a wide range of domains, including supply chain optimization, traffic congestion prediction, predictive maintenance in industrial systems, personalized recommendation systems, clinical event forecasting in healthcare, and modeling dynamic communities in social networks.

However, despite the importance of accurate element-level predictions, existing methods face significant computational challenges with large temporal sets. Attention-based mechanisms typically scale quadratically with sequence length, while many graph-based approaches have quadratic or worse complexity relative to the number of elements. These computational constraints limit applicability to real-world scenarios where both the universe of elements and the sequence length can be substantial. For dynamic environments requiring frequent updates, such as real-time recommendation systems or network monitoring, these performance limitations become particularly problematic.

In this paper, we propose an architecture called **PIETSP** for temporal set prediction that reduces computational complexity to $\mathcal{O}(N(KD + D^2) + |E|D)$, where $N$ is the number of distinct elements in a sequence of sets, $K$ is the maximum sequence length, $D$ is the embedding dimension, and $|E|$ is the domain of all possible elements (vocabulary size). This represents a significant improvement over conventional attention or graph based approaches which typically incur quadratic complexities in $N$ or $K$. The proposed architecture offers both scalability and accuracy in predicting set membership at future time points. Our contributions include:

Submitted to 39th Conference on Neural Information Processing Systems (NeurIPS 2025). Do not distribute.

- A mathematically principled formulation of temporal set prediction that integrates element features and their temporal dynamics in a joint representation, allowing for more accurate and efficient predictions.

- A novel algorithm that achieves linear scaling with respect to both sequence length and number of distinct elements independently, thus enabling the processing of large-scale datasets in a computationally efficient manner.

- Comprehensive empirical evaluation on publicly available datasets, demonstrating that our approach offers comparable or superior performance to existing state-of-the-art methods while significantly reducing computational requirements.

## 2 Related Work

**Temporal Set Prediction.** Temporal Set Prediction (TSP) is a generalization of sequence prediction that models the evolution of sequences of unordered sets rather than sequences of individual elements. Several baselines have been proposed for this task. Sets2Sets [1] formulates it as sequential set-to-set learning using an RNN encoder-decoder with set attention and repeated element modules; however, its recurrent structure limits parallelism and slows training. DNNTSP [2] extends this by modeling dynamic co-occurrence graphs using GCNs [3], incorporating temporal attention and gated fusion to capture sequence dynamics, albeit with increased memory and compute costs due to graph construction. SFCNTSP [4] mitigates these issues through a lightweight architecture with permutation invariant and equivariant layers, achieving faster inference and fewer parameters, though scalability remains a challenge on large datasets.

**Next Basket and Set Prediction.** TSP is closely aligned with next-basket recommendation, where the goal is to predict the next set of items a user will interact with. Early models such as FPMC [5] combine matrix factorization with Markov Chains to model user preferences and item transitions. While efficient, FPMC lacks the ability to model inter-item dependencies within a basket and does not generalize well to cold or rare items. Additionally, models such as SHAN [6] and SASRec [7] introduced self-attention mechanisms [8] to this task, but their adaptation to set sequences is limited, as attention is inherently order-sensitive and does not handle set permutation-invariance without explicit modification.

**Multiset Modeling and Repetition.** A defining feature of TSP, and a gap in traditional sequential models, is the ability to handle repeated elements—important in domains like healthcare (e.g., recurring diagnoses, lab tests) and e-commerce (e.g., repeated purchases). Sets2Sets and DNNTSP directly model repetition via frequency-aware loss functions or modules that attend to past occurrences. However, these often assume hard duplication counts rather than modeling repetition as a stochastic process.

**Sequence-to-Sequence and Set Modeling.** TSP extends the classic sequence-to-sequence (seq2seq) framework popularized in NLP [9, 10], but demands adaptations for sets due to their unordered and variable-length nature. Key inspirations come from DeepSets [11], which introduced permutation-invariant functions for effective set representation. Building on this foundation, Set Transformers [12] leverage attention mechanisms specifically designed for set inputs and outputs. SetVAE [13] further advances this line of research by enabling generative modeling of unordered outputs with improved computational efficiency.

While standard Transformer-based approaches often suffer from quadratic complexity in set size, SetVAE employs architectural innovations to mitigate this limitation. However, challenges remain in capturing sequential dependencies when these models are applied to problems requiring both set-level operations and order-sensitive processing.

## 3 Problem Formulation

Let $S(t_i) \subseteq E$ represent the set of all elements present at time $t_i$, where $E$ is the domain of possible elements. Let $\mathcal{S} = [S(t_1), S(t_2), \ldots, S(t_T)]$ be the sequence of sets across $T$ time steps, where $1 \leq T \leq K$ and $K = \max\{|\mathcal{S}_j| : \mathcal{S}_j \in \mathcal{D}\}$ is the maximum set sequence length across the dataset $\mathcal{D}$. Let $U = \bigcup_{i=1}^{T} S(t_i)$ be the universe of all distinct elements that appear in any set in the sequence

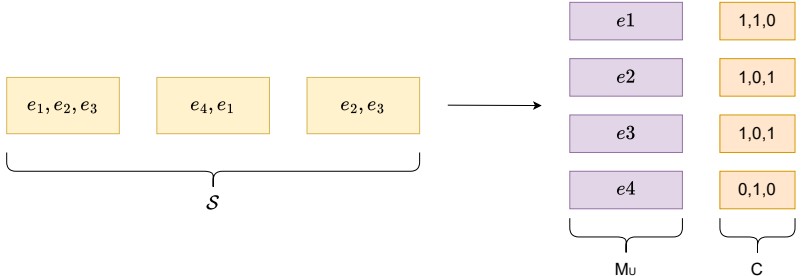

Figure 1: The process of $M_U$ and $C$ construction

$\mathcal{S}$. We can enumerate the elements in this universe as $U = \{e_1, e_2, \ldots, e_N\}$, where $N = |U|$ is the total number of unique elements in $\mathcal{S}$.

Let $M \in \mathbb{R}^{|E| \times D}$ be the embedding matrix for the entire domain $E$, where each row represents the $D$-dimensional embedding vector for an element in the domain. Let $M_U \in \mathbb{R}^{N \times D}$ be the embedding matrix for elements in $U$, where $M_U$ is the submatrix of $M$ containing only the rows corresponding to elements in $U$. We show the construction of $M_U$ in Figure 1.

Given this formulation, our objective is to predict the future set membership $S(t_{T+1})$ by analyzing patterns in the sequence history $\mathcal{S}$. Specifically, we aim to develop models that can accurately forecast element membership in the set at time $T + 1$, based on the structural patterns identified in the evolution history of $\mathcal{S}$.

# 4 Methodology

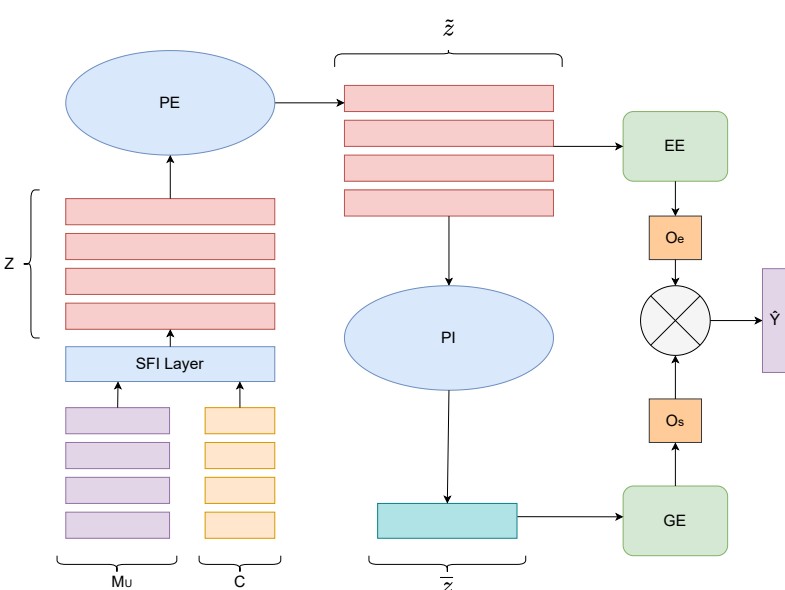

Figure 2: Architecture of the proposed model. The input set elements and their sequence features are combined using the Sequence Feature Integration (SFI) layer and passed through permutation-equivariant (PE) and permutation-invariant (PI) blocks, followed by combining the element-wise (EE) and global evaluators(GE) producing the final prediction.

We first give a brief overview of the sequence of operations below, as shown in Figure 2 and elaborate each operation in the subsections. In order to predict the future set membership $S(t_{T+1})$, we start by

integrating the embeddings $M_U$ of the unique elements $N$ with a sequence feature $C$ (illustrated in Figure 1) by passing it through **Sequence Feature Integration** (**SFI**) layer. The resulting output $Z$ of $N$ elements is passed to a **permutation equivariant layer** (**PE**). This enriches the representation of the set elements with a global aggregated context. The updated representation $\widetilde{Z}$ captures the integrated sequence-element relationship and serves as input to two separate computational branches as shown in Figure 2. The branch on the right, called **Element Evaluator** (**EE**), generates scores for each of the updated $N$ elements, collectively referred to as $O_e$.

Meanwhile, the branch extending downwards passes $\widetilde{Z}$ through a **permutation invariant layer** (**PI**). PI transforms the updated set of $N$ elements $\widetilde{Z}$ into a single representation $\overline{Z}$. Note that $\overline{Z}$ is the representation of an entire sequence of sets. $\overline{Z}$ is then passed through **Global Evaluator** (**GE**) to get scores $Os$ for the entire domain $E$. $O_e$, the scores for updated $N$ elements $\widetilde{Z}$ and $Os$, the global scores for the entire domain $E$, are then fused to get the final logits $\hat{Y}$. Since our method applies a Permutation Invariant operation (PI) following a Permutation Equivariant operation (PE) for Temporal Set Prediction (TSP), we name have named our proposed approach **PIETSP**.

## 4.1 Sequence Feature Integration

Our approach differs from related methods that typically process set elements for each time step in isolation. Although conventional techniques distribute the $N$ distinct sequence elements across the $K$ time steps using weight-sharing mechanisms for separate processing, our method integrates comprehensive sequence information for each of the $N$ elements. We utilize a sequential representation $C \in \mathbb{R}^{N \times Q}$ that encodes relationships between elements and sequences. The matrices $M_U$ and $C$ are integrated via the Sequence Feature Integration (**SFI**) function, as outlined in (1), resulting in matrix $Z$. This matrix $Z$ combines element-specific features with sequential information which may include membership patterns, temporal embeddings, or count-based representations.

$$Z = \text{SFI}(M_U, C) \in \mathbb{R}^{N \times F} \tag{1}$$

For our implementation, we define $C$ as a multi-hot representation $C \in \{0, 1\}^{N \times K}$, where:

$$C[i, j] = \begin{cases} 1 & \text{if element } e_i \in S(t_j) \\ 0 & \text{if element } e_i \notin S(t_j) \end{cases} \tag{2}$$

In our implementation, the matrix $C$ efficiently encodes the binary membership relationships between elements and sequences, enabling us to mathematically model these relationships with each row corresponding to an element and each column representing a sequence. The resulting structure preserves the distributional patterns of elements across sequences, forming the basic for subsequent operations in our method. We use simple concatenation for SFI. This results in $Z$ of shape $N \times (K + D)$. While the original data consists of a sequence of sets, transforming it into the matrix $Z$ enables efficient neural processing using standard operations, while preserving the underlying set semantics through permutation-aware design.

## 4.2 Integrated Element-Sequence Relationship Learning

To effectively capture the interplay between individual elements and the sequences they participate in, we apply a permutation equivariant transformation to the matrix $Z$. A function $f$ is *permutation equivariant* if permuting its inputs results in an equivalent permutation of its outputs. Formally, for any permutation $\pi$ and input list $X = [x_1, x_2, ..., x_n]$, a permutation equivariant function satisfies:

$$f([x_{\pi(1)}, x_{\pi(2)}, ..., x_{\pi(n)}]) = [f(X)]_\pi$$

This property ensures that our model respects the structure of the input while allowing meaningful transformations of individual elements based on the collective context. We pass $Z \in \mathbb{R}^{N \times (K+D)}$ through a permutation equivariant layer to obtain $\widetilde{Z} \in \mathbb{R}^{N \times d'}$, as shown in Equation (4):

$$\widetilde{Z} = \text{PE}(Z) \in \mathbb{R}^{N \times d'} \tag{3}$$

In our implementation, we use a mean permutation equivariant layer, defined as:

$$\widetilde{Z} = \text{ELU}\left(ZW_g + b_g - \frac{1}{N}\sum_{i=1}^{N}(Z_i W_\ell)\right) \in \mathbb{R}^{N \times d'} \tag{4}$$

where $W_g \in \mathbb{R}^{(K+D) \times d'}$ and $W_\ell \in \mathbb{R}^{(K+D) \times d'}$ are learnable weight matrices, and $b_g \in \mathbb{R}^{d'}$ is a learnable bias vector. The ELU activation is applied element-wise to the entire result to introduce smooth, non-linear transformations.

To reduce computational complexity, we set the output dimension $d' = D$, thereby projecting $Z \in \mathbb{R}^{N \times (K+D)}$ into $\mathbb{R}^{N \times D}$. This avoids the quadratic cost of a full $(K + D) \times (K + D)$ transformation and instead yields a more efficient $\mathcal{O}(N(K + D)D)$ complexity, which is *linear* in the number of time steps $K$ when $D$ is fixed. The resulting output matrix $\widetilde{Z} \in \mathbb{R}^{N \times D}$ is then used in subsequent stages of the model.

## 4.3 Element Evaluator

Following the permutation equivariant transformation, we apply an element-wise evaluator (EE) to the enriched representations $\widetilde{Z}$ in order to compute scalar relevance scores for each element. Specifically, the evaluator produces one score per element as defined in Equation (5):

$$O_e = \text{EE}(\widetilde{Z}) \in \mathbb{R}^N \tag{5}$$

In our implementation, the EE is instantiated as a two-layer Multi-Layer Perceptron (MLP) with a ReLU activation in between. This setup allows the model to assess each element's importance based on both its intrinsic characteristics and its contextual role within the sequence.

Since the evaluator processes elements independently, the permutation equivariant structure established earlier is preserved. The resulting scores $O_e \in \mathbb{R}^N$ quantify each element's contextual relevance and serve as intermediate signals, which will later be merged with complementary scores to inform the final prediction.

## 4.4 Sequence Set Representation

To derive a global representation of the input sequence, we aggregate information from the enriched element representations in a way that is invariant to element order. This summary will later be used to complement the element-level scores for final prediction.

A function $f$ is *permutation invariant* if its output remains unchanged regardless of the ordering of its input elements. Formally, for any permutation $\pi$ and input list $X = [x_1, x_2, \dots, x_n]$, a permutation invariant function satisfies:

$$f([x_{\pi(1)}, x_{\pi(2)}, \dots, x_{\pi(n)}]) = f([x_1, x_2, \dots, x_n])$$

This property ensures that the model produces consistent outputs for a given collection of elements, independent of their order. In our method, we apply a permutation invariant operation to the enriched element representations $\widetilde{Z} \in \mathbb{R}^{N \times D}$ to obtain a global sequence-level summary vector:

$$\overline{Z} = \text{PI}(\widetilde{Z}) \in \mathbb{R}^{1 \times D} \tag{6}$$

Specifically, we use a sum pooling operation followed by a Multi-Layer Perceptron (MLP) consisting of two hidden layers with ELU activations and a final output layer:

$$\overline{Z} = \text{MLP}\left(\sum_{i=1}^{N} \widetilde{Z}_i\right) \tag{7}$$

The resulting summary $\overline{Z}$ encapsulates global sequence-level context that informs the final element selection.

## 4.5 Global Evaluator

We perform global scoring using a global evaluator (GE), which computes relevance scores for all elements in the domain $E$, as shown in Equation (8).

$$O_s = \text{GE}(\overline{Z}) \in \mathbb{R}^{|E|} \tag{8}$$

This mechanism captures the relationship between the global sequence set representation $\overline{Z}$ (which encodes the entire sequence) and each candidate element in the domain $E$. The global evaluator could take various forms, such as dot product similarity, concatenation followed by an MLP, or other scoring functions. In our implementation, we specifically use a dot product formulation to calculate the scores, measuring the similarity between each element embedding in $M$ and the global sequence representation $\overline{Z}$, as shown in Equation (9).

$$O_s = M \cdot \overline{Z}^\top \tag{9}$$

This approach produces a score for each element in $E$, indicating its relevance according to the global sequence context.

## 4.6 Score Fusion

To effectively combine global context information with element-specific sequential patterns, we implement a score fusion mechanism. This approach integrates global scores for all domain elements with element-level scores from the set sequence. We introduce learnable parameter vectors $\boldsymbol{\alpha}, \boldsymbol{\beta} \in \mathbb{R}^{|E|}$ to weight each information source. The global scores $O_s \in \mathbb{R}^{|E|}$ for all elements in domain $E$ from Equation (8) and the element-level scores $O_e \in \mathbb{R}^N$ from the set sequence as defined in Equation (5) serve as inputs to our score fusion mechanism.

Let $I : \{1, \ldots, N\} \rightarrow \{1, \ldots, |E|\}$ be a one-to-one mapping function that maps each element index $i$ in the set sequence to its unique corresponding index $j$ in the domain $E$. Let $D_{\text{seq}} \subset \{1, \ldots, |E|\}$ be the set of domain indices that are mapped from the set sequence, i.e.,

$$D_{\text{seq}} = \{j \in \{1, \ldots, |E|\} : \exists i \in \{1, \ldots, N\}, I(i) = j\}$$

The final logit output $\hat{Y} \in \mathbb{R}^{|E|}$ is computed as:

$$\hat{Y}_j = \begin{cases} \alpha_j \cdot (O_s)_j + \beta_j \cdot (O_e)_i & \text{if } j \in D_{\text{seq}} \text{ where } I(i) = j \\ \alpha_j \cdot (O_s)_j & \text{otherwise} \end{cases} \tag{10}$$

This formulation ensures that all elements receive a score based on global context (weighted by $\alpha_j$), while only elements present in the set sequence receive an additional contribution from their element-level scores (weighted by $\beta_j$). It allows the model to adaptively balance local sequential patterns (captured by $O_e$) with global context information (captured by $O_s$) when determining the likelihood of each element appearing in the next set.

## 4.7 Model Training Process

Our model is trained with a batch size of 64. Sequences are zero-padded at the beginning in the multi-hot sequence feature representation $C$, as formalized in Equation (2), and $K$ is set to 19. We use an embedding dimension of 32 and optimize using Adam with a learning rate of 0.001 and weight decay of 0.01. The learning rate follows a cosine decay schedule. We train for 100 epochs with early stopping (patience=10). Given that the prediction of next-period item sets constitutes a multi-label classification problem, we implement a binary cross-entropy loss function.

## 4.8 Model Complexity Analysis

**Time Complexity:** Our model demonstrates efficient computational scaling across its components. The time complexity for element relationship learning using PE is $\mathcal{O}(N(K + D) \cdot D)$, while creating the set sequence embedding via PI requires $\mathcal{O}(ND^2)$ operations. Scoring set elements via EE contributes to an additional $\mathcal{O}(ND^2)$ complexity, and global scoring of all elements in domain $E$ using GE adds $\mathcal{O}(|E|D)$ operations. Finally, the score fusion layer adds $\mathcal{O}(|E|)$ operations. The total time complexity can be expressed as:

$$\mathcal{O}(N(K + D) \cdot D + ND^2 + ND^2 + |E|D + |E|)$$

This simplifies to:

$$\mathcal{O}(N(KD + D^2) + |E|D)$$

| Datasets | #sets | #users | #elements | #E/S | #S/U |
|----------|-------|--------|-----------|------|------|
| TaFeng | 73,355 | 9,841 | 4,935 | 5.41 | 7.45 |
| DC | 42,905 | 9,010 | 217 | 1.52 | 4.76 |
| TaoBao | 628,618 | 113,347 | 689 | 1.10 | 5.55 |
| TMS | 243,394 | 15,726 | 1,565 | 2.19 | 15.48 |

Table 1: Dataset statistics

This formulation confirms our model's efficient scaling with respect to the number of elements $N$, maximum sequence length $K$, embedding dimensionality $D$, and vocabulary size $|E|$. Our model offers computational advantages compared to existing approaches, achieving linear scaling with respect to both the number of elements $N$ and maximum sequence length $K$.

This efficiency, coupled with time complexity that remains independent of the number of layers, represents a significant improvement over related methods that either scale quadratically with $N$ or $K$, or have complexity that grows linearly with the number of layers in the model.

**Space Complexity:** The primary memory cost arises from the element embeddings with complexity $O(|E|D)$, which is standard across similar methods. Beyond this, PIETSP introduces the PE which is $O((K + D)D)$, PI is $O(D^2)$, an element scorer with $O(D^2)$ complexity and score fusion with $O(|E|)$. This leads to a total space cost of:

$$\mathcal{O}(|E|D + (K + D)D + D^2 + D^2 + |E|)$$

This simplifies to:

$$\mathcal{O}(D(|E| + K + D))$$

# 5 Experiments

This section details the experimental setup used to evaluate our approach. We describe the datasets employed in our study, followed by the baseline methods used for comparison. We use three standard metrics for top-$k$ recommendation: Recall@$k$, nDCG@$k$, and PHR@$k$. Recall@$k$ measures the proportion of relevant items retrieved, while nDCG@$k$ captures both relevance and ranking quality. PHR@$k$ indicates the fraction of users for whom at least one relevant item appears in the top-$k$ predictions. We report results at multiple cutoffs to provide a comprehensive evaluation.

## 5.1 Datasets

We evaluate our model on four publicly available datasets commonly used in temporal set prediction and next basket recommendation tasks: **TaFeng**, **Dunnhumby-Carbo (DC)**, **TaoBao**, and **Tags-Math-Sx (TMS)**. Each dataset records user behaviors over time as sequences of sets, where each set contains the items associated with a user's interaction at a particular timestamp. The last set in the sequence is used as the label. We discuss some relevant statistics for the datasets used in Table 1 where #E/S denotes the average number of elements in each set, #S/U represents the average number of sets for each user. Our datasets and the train,validation and test splits have been sourced from https://github.com/yule-BUAA/DNNTSP/tree/master/data.

## 5.2 Baseline Methods

To evaluate the effectiveness of our proposed approach, we compare it against three state-of-the-art models designed for temporal set prediction: **Sets2Sets**, **DNNTSP**, and **SFCNTSP**. These methods represent diverse modeling strategies, including recurrent, graph-based, and fully connected architectures.

**Sets2Sets** [1]. Sets2Sets formulates temporal set prediction as a sequential sets-to-sequential sets learning problem. It employs an encoder-decoder architecture built on recurrent neural networks (RNNs), where each input set is embedded via a set-level embedding mechanism, and the sequence is modeled with a decoder using set-based attention. It also incorporates a repeated-elements module to capture frequent historical patterns and a custom objective function to address label imbalance and label correlation.

| Dataset (p95 Set Size) | Method | Recall | | | | NDCG | | | | PHR | | | |
|---|---|---|---|---|---|---|---|---|---|---|---|---|---|
| | | @1 | @2 | @5 | @10 | @1 | @2 | @5 | @10 | @1 | @2 | @5 | @10 |
| Tafeng (15) | Sets2Sets | 0.0302 | 0.0477 | 0.0832 | 0.1264 | 0.0767 | 0.0754 | 0.0820 | 0.0965 | 0.0767 | 0.1254 | 0.2158 | 0.3296 |
| | DNNTSP | 0.0448 | 0.0694 | 0.1140 | 0.1692 | 0.1422 | 0.1318 | 0.1293 | 0.1436 | 0.1422 | 0.2240 | 0.3509 | 0.4708 |
| | SFCNTSP | 0.0471 | 0.0728 | 0.1126 | 0.1674 | 0.1503 | 0.1378 | 0.1303 | 0.1437 | 0.1503 | 0.2336 | 0.3545 | 0.4703 |
| | PIETSP | **0.0515** | **0.0833** | **0.1300** | **0.1866** | **0.1778** | **0.1635** | **0.1531** | **0.1650** | **0.1778** | **0.2702** | **0.3885** | **0.4967** |
| DC (3) | Sets2Sets | 0.1276 | 0.2311 | 0.3825 | 0.4259 | 0.1776 | 0.2185 | 0.2883 | 0.3041 | 0.1775 | 0.3123 | 0.4786 | 0.5219 |
| | DNNTSP | 0.1480 | 0.2424 | 0.3924 | 0.4609 | 0.2047 | 0.2346 | 0.3035 | 0.3282 | 0.2047 | 0.3256 | 0.4870 | 0.5528 |
| | SFCNTSP | 0.1581 | 0.2457 | 0.3879 | 0.4585 | 0.2174 | 0.2421 | 0.3063 | 0.3330 | 0.2174 | 0.3295 | 0.4836 | 0.5552 |
| | PIETSP | **0.1811** | **0.2632** | **0.3983** | **0.4615** | **0.2514** | **0.2641** | **0.3235** | **0.3463** | **0.2514** | **0.3489** | **0.4958** | **0.5613** |
| TaoBao (2) | Sets2Sets | 0.0019 | 0.0398 | 0.0985 | 0.1743 | 0.0019 | 0.0260 | 0.0521 | 0.0767 | 0.0019 | 0.0409 | 0.1015 | 0.1787 |
| | DNNTSP | 0.0786 | 0.1394 | 0.2289 | 0.3032 | 0.0812 | 0.1183 | 0.1590 | 0.1831 | 0.0812 | 0.1434 | 0.2337 | 0.3093 |
| | SFCNTSP | 0.1003 | 0.1577 | 0.2355 | **0.3103** | 0.1037 | 0.1383 | 0.1766 | 0.1952 | 0.1037 | 0.1619 | 0.2402 | **0.3165** |
| | PIETSP | **0.1116** | **0.1613** | **0.2364** | 0.3059 | **0.1155** | **0.1448** | **0.1781** | **0.2012** | **0.1155** | **0.1656** | **0.2410** | 0.3108 |
| TMS (4) | Sets2Sets | **0.2055** | 0.2782 | 0.3589 | 0.4423 | **0.3846** | 0.3408 | 0.3455 | 0.3743 | **0.3846** | 0.4637 | 0.5645 | 0.6557 |
| | DNNTSP | 0.1248 | 0.2131 | 0.3566 | 0.4691 | 0.2616 | 0.2561 | 0.3000 | 0.3453 | 0.2616 | 0.3789 | 0.5633 | 0.6844 |
| | SFCNTSP | 0.1681 | 0.2655 | 0.3940 | 0.4960 | 0.3210 | 0.3133 | 0.3490 | 0.3924 | 0.3210 | 0.4469 | 0.5995 | 0.7044 |
| | PIETSP | 0.1930 | **0.2852** | **0.4074** | **0.4982** | 0.3620 | **0.3412** | **0.3713** | **0.4075** | 0.3620 | **0.4638** | **0.6068** | **0.7092** |

Table 2: Performance comparison on four datasets. The best results per metric are in bold.

**DNNTSP** [2]. DNNTSP is a deep neural network architecture that captures both intra-set and inter-set dependencies through graph-based modeling. It constructs dynamic co-occurrence graphs over elements within each set and applies weighted graph convolutional layers to model relationships. Additionally, it uses an attention-based temporal module to capture sequence dynamics and a gated fusion mechanism to integrate static and dynamic element representations for improved predictive accuracy.

**SFCNTSP** [4]. SFCNTSP proposes a lightweight and efficient architecture based entirely on simplified fully connected networks (SFCNs), eliminating non-linear activations and complex modules such as RNNs or attention. It captures inter-set temporal dependencies, intra-set element relationships, and channel-wise correlations through permutation-invariant and permutation-equivariant operations. Despite its simplicity, it achieves competitive performance while significantly reducing computational and memory costs.

# 6 Results

In this section, we present a comprehensive evaluation of our proposed method. We begin with a performance comparison against state-of-the-art baselines to demonstrate the effectiveness of our approach. Next, we assess the efficiency of the model in terms of model training and inference. Finally, we conduct an ablation study to analyze the contribution of different components in our architecture, which is described in the Appendix section A.1.

## 6.1 Performance Comparison

We evaluate the performance of our proposed method against several strong baselines across four benchmark datasets. We select cut-off $k \in \{1, 2, 5, 10\}$ to span single-item through longer-list predictions relative to each dataset's 95[th]-percentile set size (p95 ranges from 2 to 15 across our benchmarks). As shown in Table 2, our model consistently outperforms all baselines on Tafeng and DC and leads on TaoBao for $k \leq 5$, with only a slight drop at $k = 10$. On TMS, Sets2Sets attains the highest performance at $k = 1$ by memorizing the most frequent next element, but PIETSP surpasses every method for $k \geq 2$. These results underscore the robustness of our approach across varying set sizes and—to our knowledge—PIETSP establishes a new state-of-the-art performance on all four benchmarks under this evaluation.

## 6.2 Model Efficiency Comparison

To evaluate the efficiency of our proposed method, we compare it with the baseline SFCNTSP model [4]. While SFCNTSP achieves competitive performance by eliminating complex modules like RNNs and attention, it still faces higher time complexity compared to our approach. This higher complexity arises from the dependence on the term $\mathcal{O}(|E|ND)$ in their adaptive fusing of user representations

| Dataset | Model | Mean Time (s) | P99 Time (s) | Samples/sec | Epochs |
|---------|-------|---------------|--------------|-------------|--------|
| TaFeng | SFCNTSP | 0.00214 | 0.00250 | 467.69 | 327 |
| | **PIETSP** | 0.00011 | 0.00017 | 9081.62 | 14 |
| DC | SFCNTSP | 0.00083 | 0.00096 | 1204.03 | 305 |
| | **PIETSP** | 0.00012 | 0.00061 | 8010.45 | 8 |
| TaoBao | SFCNTSP | 0.00091 | 0.00109 | 1097.26 | 446 |
| | **PIETSP** | 0.00010 | 0.00013 | 9799.39 | 8 |
| TMS | SFCNTSP | 0.00238 | 0.00258 | 419.97 | 313 |
| | **PIETSP** | 0.00012 | 0.00064 | 8628.95 | 11 |

Table 3: Inference speed and training efficiency comparison of SFCNTSP and PIETSP.

layer. Given that $|E| \gg N$, this dependence on the element domain size $|E|$ can lead to significant computational costs, particularly as the number of elements grows. In contrast, our method achieves a lower complexity of $\mathcal{O}(|E|D)$.

We focus on SFCNTSP for this comparison because it shares a similar goal of reducing computational cost while achieving efficient performance. However, PIETSP reduces time complexity further, providing both lower latency and higher throughput in large-scale settings.

Table 3 presents a detailed comparison of inference performance using mean sample time, 99th percentile latency (P99), and throughput (samples per second). We tested the models on Nvidia T4 GPU, across 100 runs with batch size 64 and embedding dimension $D$ of 32. Across all datasets, PIETSP consistently outperforms SFCNTSP, exhibiting lower latency and significantly higher throughput. These results highlight the practical advantages of our model in time-sensitive and large-scale deployments.

Table 3 also shows the number of epochs required for each model to converge. Our proposed model, PIETSP, achieves convergence significantly faster, requiring substantially fewer training epochs than SFCNTSP across all datasets. This demonstrates its training efficiency and potential for rapid iteration in real-world systems.

# 7 Conclusion

We propose PIETSP, a scalable and permutation-aware model for temporal set prediction that achieves linear time complexity with respect to both sequence length and element count. By integrating permutation-equivariant and permutation-invariant operations, PIETSP enables efficient modeling of evolving sets and offers significant improvements in inference speed and training efficiency.

Empirical results across four public benchmarks show that PIETSP achieves comparable or superior performance to existing state-of-the-art models while requiring significantly fewer computational resources.

# 8 Limitations and Future Work

While efficient, the current architecture may under-capture fine-grained inter-element dependencies. Enhancing expressiveness via more advanced attention mechanisms (e.g., Set Transformers) is a promising direction. Additionally, the model operates on a fixed-length temporal window, which may limit its effectiveness on long-range dependencies. Lastly, our work does not explore fairness or uncertainty estimation, both of which are important considerations for high-stakes applications and future work.

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

## A  Technical Appendices and Supplementary Material

### A.1  Ablation Study

To better understand the contribution of each component in our architecture, we conduct an ablation study. We compare the full PIETSP model with two variants: PIETSP-EE and PIETSP-GE, where we remove the element evaluator EE and global evaluator GE modules respectively. We test the variants on the Tafeng dataset. As shown in Table 4, removing either component leads to a noticeable drop in performance, confirming the importance of both elements in capturing temporal and contextual patterns effectively. The full model consistently outperforms both ablated variants, demonstrating that the synergy between EE and GE is crucial to the overall effectiveness of the proposed approach.

### A.2  Statistical Variability in Experimental Results

We present the variability in the experimental results of PIETSP across various metrics, reporting the mean along with two standard deviations as shown in Table 5

| Dataset | Method | Recall | | | | NDCG | | | | PHR | | | |
|---------|--------|--------|--------|--------|--------|--------|--------|--------|--------|--------|--------|--------|--------|
| | | @1 | @2 | @5 | @10 | @1 | @2 | @5 | @10 | @1 | @2 | @5 | @10 |
| | PIETSP-GE | 0.0043 | 0.0050 | 0.0058 | 0.0060 | 0.0218 | 0.0156 | 0.0107 | 0.0090 | 0.0218 | 0.0239 | 0.0274 | 0.0284 |
| Tafeng | PIETSP-EE | 0.0452 | 0.0669 | 0.1004 | 0.1333 | 0.1427 | 0.1178 | 0.1098 | 0.1163 | 0.1427 | 0.1910 | 0.2712 | 0.3393 |
| | **PIETSP** | **0.0515** | **0.0833** | **0.1300** | **0.1866** | **0.1778** | **0.1635** | **0.1531** | **0.1650** | **0.1778** | **0.2702** | **0.3885** | **0.4967** |

Table 4: Ablation study on the Tafeng dataset. PIETSP-GE: global evaluator removed; PIETSP-EE: element evaluator removed. Best results per metric are in bold.

| Dataset | Metric | @1 | @2 | @5 | @10 |
|---------|--------|------|------|------|------|
| TaFeng | PHR | 0.1756 ± 0.0068 | 0.2661 ± 0.0142 | 0.3926 ± 0.0089 | 0.5009 ± 0.0148 |
| | nDCG | 0.1756 ± 0.0068 | 0.1612 ± 0.0074 | 0.1530 ± 0.0041 | 0.1655 ± 0.0042 |
| | Recall | 0.0516 ± 0.0022 | 0.0830 ± 0.0042 | 0.1316 ± 0.0070 | 0.1891 ± 0.0108 |
| DC | PHR | 0.2466 ± 0.0048 | 0.3485 ± 0.0038 | 0.4958 ± 0.0021 | 0.5617 ± 0.0028 |
| | nDCG | 0.2466 ± 0.0048 | 0.2633 ± 0.0033 | 0.3227 ± 0.0018 | 0.3457 ± 0.0016 |
| | Recall | 0.1779 ± 0.0034 | 0.2645 ± 0.0030 | 0.3984 ± 0.0015 | 0.4624 ± 0.0020 |
| TaoBao | PHR | 0.1156 ± 0.0017 | 0.1672 ± 0.0021 | 0.2410 ± 0.0016 | 0.3111 ± 0.0014 |
| | nDCG | 0.1156 ± 0.0017 | 0.1459 ± 0.0016 | 0.1787 ± 0.0010 | 0.2010 ± 0.0010 |
| | Recall | 0.1118 ± 0.0015 | 0.1630 ± 0.0021 | 0.2364 ± 0.0017 | 0.3052 ± 0.0013 |
| TMS | PHR | 0.3616 ± 0.0036 | 0.4672 ± 0.0049 | 0.6073 ± 0.0017 | 0.7048 ± 0.0047 |
| | nDCG | 0.3616 ± 0.0036 | 0.3418 ± 0.0026 | 0.3714 ± 0.0014 | 0.4073 ± 0.0015 |
| | Recall | 0.1927 ± 0.0020 | 0.2861 ± 0.0026 | 0.4080 ± 0.0020 | 0.4956 ± 0.0030 |

Table 5: Performance of our proposed model on the datasets. Each value is reported as mean ± 2xstd.

