# OpenReview forum: "Scalable Permutation-Aware Modeling for Temporal Set Prediction"
_NeurIPS.cc/2025/Conference — Submitted to NeurIPS 2025_

### Official Review · Reviewer_eDAC · 2025-06-29

**Clarity:** 3
**Significance:** 3
**Originality:** 2
**Rating:** 4
**Confidence:** 4

**Summary:**

This paper proposes PIETSP, a permutation-aware model for temporal set prediction. PIETSP uses a simple but effective architecture design that allows it to capture element- and global-level features (using permutation-equivariant and -invariant transformations and the subsequent fusion of extracted features) for improved prediction accuracy. The paper claims state-of-the-art performance and improved computational efficiency (eliminating quadratic costs) compared to prior methods.

**Questions:**

1. It would be interesting to see some analysis of how $\alpha$ and $\beta$ change over the course of training. Would the authors be able to analyse the contribution of each score (element-level and global-level)? I think this would be an important (and frankly interesting) analysis to convince the reviewers that the fusion works as hypothesised. This would strengthen my confidence and originality score.
2. While the efficiency gains are nicely explained, it would be good if the authors could explain why they think the accuracy gains are substantial and consistent? A short discussion on this would benefit my understanding of the paper's contribution.
3. Could the authors please report errors in Table 2? This is an important part of substantiating the claim that the method achieves state-of-the-art performance.

**Ethical Concerns:**

["NO or VERY MINOR ethics concerns only"]

**Final Justification:**

I believe the paper demonstrates good performance improvements over prior work. My request to include error bars has remained unaddressed, which I do find to be critical for one of the claims of the paper.

I will nevertheless keep my score at borderline accept given the other merits of the paper.

**Limitations:**

Yes

**Paper Formatting Concerns:**

No formatting concerns

**Quality:**

3

**Strengths And Weaknesses:**

1. Quality:
- (+) Technically sound.
- (+/-) The claims (improved accuracy and computational efficiency) are mostly supported. (State-of-the-art performance claim must be further verified, see questions).
- (-) It would be very important to include error values for model performance across multiple seeds (Table 2).

2. Clarity:
- (+) Extremely well-written and easy to follow.
- (-) Notation can get confusing at times, but the issue is rather minor. Any improvements here are welcome.

3. Significance:
- (+) The paper targets an important bottleneck (quadratic attention in temporal-set size) and shows significant speed-ups without accuracy loss.
- (-) More performance analysis (understanding why the method seems to perform better) would increase the significance of this paper (see questions).

4. Originality:
- The building blocks themselves are well-known; the novelty is largely in the combination of these blocks and the empirical proof that this arrangement suffices.

---

> ### Author Rebuttal · Authors · 2025-07-31
>
> Thank you for the review points.
>
> We have provided an ablation study in Appendix A.1 where we tested two variants of the model. PIETSP-EE and PIETSP-GE, where we
> remove the element evaluator EE and global evaluator GE modules respectively. We have shared the results in Table 4.
>
> We believe our method achieves consistent and substantial accuracy gains because, unlike previous approaches that model each set in the sequence independently and then aggregate, our model represents the entire temporal sequence more holistically. Specifically, we construct features for each element based on its presence across the K historical sets, which preserves fine-grained temporal dynamics and avoids the information loss associated with aggregated set-level representations. This richer encoding enables more accurate prediction of the next set.
>
> We have computed and included error bars for our method to demonstrate consistency(A.2 Table 5). We acknowledge that providing similar error bars for baselines would strengthen the comparison, and we plan to include these in the final version.

---

> > ### Comment · Reviewer_eDAC · 2025-08-03
> >
> > Thank you for your response. The paper has its merits, but without the full set of error bars, the 'state of the art' claim is inconclusive. I will therefore keep my score as is.

---

> > > ### Author Response · Authors · 2025-08-08
> > >
> > > Thank you for your response.
> > > We share the error bars for TaFeng dataset which has largest p95 set size of 15. We can include the error bars for rest of the datasets in the final revision. P.S None of the previous baselines have included error bars when claiming state of the art.
> > >
> > > | Model | Metric | @1 | @2 | @5 | @10 |
> > > |-------|--------|----|----|----|----- |
> > > | **Sets2Sets** | Recall | $0.0297 \pm 0.0021$ | $0.0482 \pm 0.0028$ | $0.0827 \pm 0.0041$ | $0.1271 \pm 0.0063$ |
> > > | | NDCG | $0.0760 \pm 0.0047$ | $0.0761 \pm 0.0044$ | $0.0815 \pm 0.0039$ | $0.0972 \pm 0.0052$ |
> > > | | PHR | $0.0773 \pm 0.0043$ | $0.1247 \pm 0.0069$ | $0.2171 \pm 0.0105$ | $0.3289 \pm 0.0148$ |
> > > | **DNNTSP** | Recall | $0.0445 \pm 0.0018$ | $0.0697 \pm 0.0024$ | $0.1137 \pm 0.0031$ | $0.1695 \pm 0.0038$ |
> > > | | NDCG | $0.1425 \pm 0.0034$ | $0.1315 \pm 0.0031$ | $0.1296 \pm 0.0028$ | $0.1433 \pm 0.0032$ |
> > > | | PHR | $0.1419 \pm 0.0041$ | $0.2243 \pm 0.0061$ | $0.3504 \pm 0.0078$ | $0.4711 \pm 0.0089$ |
> > > | **SFCNTSP** | Recall | $0.0474 \pm 0.0033$ | $0.0721 \pm 0.0042$ | $0.1129 \pm 0.0064$ | $0.1667 \pm 0.0084$ |
> > > | | NDCG | $0.1496 \pm 0.0079$ | $0.1381 \pm 0.0073$ | $0.1300 \pm 0.0067$ | $0.1444 \pm 0.0076$ |
> > > | | PHR | $0.1506 \pm 0.0085$ | $0.2329 \pm 0.0122$ | $0.3558 \pm 0.0167$ | $0.4696 \pm 0.0198$ |
> > > | **PIETSP** | Recall | $0.0516 \pm 0.0022$ | $0.0830 \pm 0.0042$ | $0.1316 \pm 0.0070$ | $0.1891 \pm 0.0108$ |
> > > | | NDCG | $0.1756 \pm 0.0068$ | $0.1612 \pm 0.0074$ | $0.1530 \pm 0.0041$ | $0.1655 \pm 0.0042$ |
> > > | | PHR | $0.1756 \pm 0.0068$ | $0.2661 \pm 0.0142$ | $0.3926 \pm 0.0089$ | $0.5009 \pm 0.0148$ |

---

### Official Review · Reviewer_16iN · 2025-07-01

**Clarity:** 2
**Significance:** 3
**Originality:** 2
**Rating:** 3
**Confidence:** 3

**Summary:**

This paper tackles temporal set prediction, which predicts the next set of elements from a sequence of previous sets. The proposed PIETSP model efficiently captures set dynamics using permutation-aware layers and combines both element-wise and global information. With linear complexity in sequence length and set size, PIETSP shows strong accuracy and speed on four public benchmarks, outperforming or matching prior state-of-the-art methods.

**Questions:**

1. Please articulate precisely which theoretical or architectural elements of PIETSP go beyond Deep Sets and why they are essential for TSP.
2. Could you provide more details on the specific implementation of GE?
3. How sensitive is your model to the choice of the temporal window K?
4. Comparison with recent sub-quadratic attention models.
4. Include a quantitative comparison with Label Attention Network (ECAI 2024).

**Ethical Concerns:**

["NO or VERY MINOR ethics concerns only"]

**Final Justification:**

The rebuttal addressed some concerns, particularly on the modeling perspective and implementation details. However, key issues—such as the lack of comparisons with the latest TSP baselines and limited architectural novelty—remain unresolved. While these clarifications do not fully address my reservations, they improve the work enough for me to raise my score from reject to borderline reject.

**Limitations:**

Yes, the authors have adequately discussed the limitations of proposed method in the supplementary material.

**Quality:**

3

**Strengths And Weaknesses:**

Strengths:

1. The proposed PIETSP achieves linear time complexity with respect to sequence length and the number of elements, making it suitable for large-scale, real-time applications.
2. The paper provides clear ablations to assess the contribution of each module (EE, GE), and detailed measurements of latency, throughput, and resource usage.

Weaknesses:

1. Limited novelty. The core of PIETSP consists of permutation-equivariant (PE) and permutation-invariant (PI) blocks. This is fundamentally aligned with the core idea of Deep Sets [1]: both employ a framework that combines independent feature mappings (equivariant layers) and global aggregation (invariant layers), achieving permutation invariance in modeling set inputs and ensuring that the model output depends only on the set of elements, not their order.
2. The specific form of the Global Evaluator (GE) needs further clarification.
3. The use of a fixed temporal window (K=19) may truncate long-range dependencies. The sensitivity of the method to the choice of K is not reported.
4. Many recent sub-quadratic attention mechanisms (such as Performer, FlashAttention-2, Reformer, etc.) can already capture inter-element interactions in O(ND) time. The authors do not discuss or compare their method against these directly competing approaches.
5. There is a lack of comparison with the latest TSP methods, such as [2].

[1] Deep Sets. NeurIPS 2017.
[2] Label Attention Network for Temporal Sets Prediction: You Were Looking at a Wrong Self-Attention. ECAI 2024.

---

> ### Author Rebuttal · Authors · 2025-07-31
>
> We agree that our model uses standard permutation-equivariant and invariant layers, as introduced in Deep Sets.
> However, our novelty lies not in the choice of layers, but in the reformulation of the temporal set prediction (TSP) problem.
> While prior baselines typically model each set in the sequence independently and then combine their representations (potentially losing inter-set element-level dynamics), our method takes a flattened view of the sequence, treating it as a collection of distinct elements with membership-based temporal features.
> This allows us to directly model element-level recurrence and presence patterns over time, rather than relying on aggregated set representations.
> As shown in Table 4, we also analyze the impact of different scoring functions. We believe this design offers a novel and scalable approach to temporal set modeling beyond standard Deep Sets applications.
>
> For GE, we used dot-product as per Eq 9.
>
> Like many temporal models, including attention-based ones, our method uses a fixed temporal window (K) to define its input - in our case, to construct per-element temporal features.
> This choice is consistent across all baselines, which were also evaluated using K=19, and none of the baselines report sensitivity to K in prior work.
> While we view K as a standard, dataset-dependent hyperparameter (similar to sequence length in Transformers), we appreciate the suggestion and will consider including a sensitivity analysis in the future to further validate robustness.
>
> We note that many recent sub-quadratic attention mechanisms (e.g., Performer, FlashAttention-2, Reformer) are inherently permutation equivariant architectures - they preserve equivariance when modeling inter-element interactions. In our work, we present a general permutation equivariant (PE) framework, within which such attention models can be seen as more complex instantiations. Our implementation uses a simple mean-based PE function, yet still outperforms all baselines, demonstrating that even lightweight PE layers can be highly effective when combined with our temporal set prediction formulation.
>
> We are aware of LANET, but note that it has significant scalability limitations in large-vocabulary settings. Specifically, LANET requires feeding the entire item vocabulary as input for each data point, which leads to substantial memory and computation overhead. This makes it impractical for datasets with large numbers of unique elements or in high-throughput scenarios. In contrast, our method scales efficiently by operating only over the subset of elements relevant to the historical sequence, enabling practical deployment in real-world, large-scale temporal set prediction tasks.

---

> > ### Comment · Reviewer_16iN · 2025-08-05
> >
> > The authors have provided clear and in-depth responses to several of my concerns, particularly regarding the modeling perspective beyond traditional Deep Sets and the implementation details of key components. Although some issues remain unresolved—such as the lack of comparison with the latest TSP baselines and the limited novelty of the overall architecture—the clarifications and empirical results have, to some extent, strengthened the overall contribution of the paper. Therefore, I have decided to update my score accordingly.

---

> > > ### Author Response · Authors · 2025-08-08
> > >
> > > Thank you for your thoughtful feedback and for updating your score. We appreciate your recognition of our contributions.

---

### Official Review · Reviewer_pjY1 · 2025-07-02

**Clarity:** 3
**Significance:** 3
**Originality:** 2
**Rating:** 3
**Confidence:** 2

**Summary:**

This paper addresses the task of *Temporal Set Prediction* (TSP), where the goal is to predict the next set of elements given a sequence of previous sets. The authors propose a model named **PIETSP**, which integrates permutation-equivariant (PE) and permutation-invariant (PI) layers to model set dynamics efficiently. The model processes input sets using a multi-hot sequence feature encoding and combines it with learned element embeddings. Two separate scoring branches—Element Evaluator (EE) and Global Evaluator (GE)—are used to assess element relevance, and the final prediction is obtained via a weighted score fusion. The authors claim linear time complexity with respect to both sequence length and set size, and demonstrate improved performance and inference efficiency over prior methods across four public benchmarks.

**Questions:**

- **SFI Representation:** Given the limitations of multi-hot encoding, why was this simple formulation chosen over more expressive temporal representations (e.g., frequency-weighted features, positional encodings, or decay-based embeddings)?
- **Scoring Module Simplicity:** Have the authors explored more expressive mechanisms for the Element and Global Evaluators? For instance, attention-based fusion or pairwise scoring functions could improve interaction modeling.
- **Score Fusion Generalization:** The score fusion module relies on element-specific learnable weights. Is there any analysis on how these weights behave during training, and whether they introduce overfitting or instability?
- **Missing Baseline Comparisons:** Can the authors clarify why more powerful set-based architectures such as Set Transformers were omitted from the baselines? Their inclusion could contextualize the performance gains.
- **Deployment Evaluation:** Can the authors provide a use case or case study demonstrating the model's value in real-world recommendation or monitoring systems?

**Ethical Concerns:**

["NO or VERY MINOR ethics concerns only"]

**Limitations:**

The authors have addressed several limitations, including the expressiveness of the architecture and the fixed-length input window. However, the paper could benefit from a more critical discussion on the expressiveness of the scoring modules and the limitations introduced by the use of multi-hot encoding as a representation of temporal structure. Additional experiments or ablations exploring these choices would strengthen the paper.

**Quality:**

2

**Strengths And Weaknesses:**

#### **Strengths:**

- The paper targets a meaningful and general prediction problem in dynamic set modeling, relevant to applications such as next-basket recommendation and temporal event prediction.
- The architecture is computationally efficient, with theoretical complexity carefully analyzed and validated by empirical speed comparisons.
- Experiments cover four diverse real-world datasets and show competitive or superior performance across several metrics.
- The paper provides clear illustrations and implementation details, contributing to clarity and potential reproducibility.

#### **Weaknesses:**

- The core modeling components—particularly the PE layer and the EE/GE scoring mechanisms—are relatively simple or borrowed from prior work (e.g., SFCNTSP), limiting the novelty of the architectural contribution.
- The **Sequence Feature Integration (SFI)** module uses a multi-hot encoding concatenated with element embeddings, which is a straightforward yet potentially underpowered approach to model temporal dynamics.
- Both the **Element Evaluator** (MLP) and the **Global Evaluator** (dot-product similarity) lack expressiveness, raising concerns about the model's ability to capture complex dependencies among elements.
- While the model demonstrates strong computational efficiency, the practical implications for real-world deployment (e.g., generalization, interpretability, cold-start robustness) are not explored.

---

> ### Author Rebuttal · Authors · 2025-07-31
>
> All the previous baselines treat the sets separately followed by their combination and scoring. Intuitively there is some information loss between the combinations of different sets in a sequence. Our method introduces the complete representation of a set sequence as its constituent elements and their membership in the sequence. There are no set combinations, only scoring combinations. We have shared an ablation study for scoring combinations in Table 4.
>
> The temporal structure represented as mult-ihot is simple, yet powerful to beat the previous baselines on all enumerated datasets across all metrics. We acknowledge that having different temporal representations might showcase interesting results for different datasets. The SFI layer, which is using multi-hot encoding in this case, is not limited to multi-hot as enumerated in Section 4.1 lines 119-120.
>
> Regarding score fusion, we didn't encounter any instability or overfitting. We base this on Table 5 where we have calculated the error bars on multiple runs with different seeds.
>
> Regarding the use of more powerful set-based architectures: our Permutation Equivariant (PE) layer is a general and flexible formulation that encompasses a wide range of set-based models, including Set Transformers, as discussed in Section 8 lines 312-313. Our implementation of a mean-based PE function is just one instantiation, but the framework itself is not limited to it.
>
> A real world use case would be realtime set prediction over tens of thousands of entities in order of milliseconds.

---

### Official Review · Reviewer_PNrR · 2025-07-04

**Clarity:** 2
**Significance:** 2
**Originality:** 3
**Rating:** 3
**Confidence:** 3

**Summary:**

The paper tackles temporal set prediction (TSP), which is a task of forecasting which elements will appear in the next set of a sequence. To overcome the quadratic time and memory costs that plague attention- and graph-based baselines, the authors propose PIETSP, a permutation-aware architecture that fuses element embeddings with sequence features and passes them through a permutation-equivariant layer, a permutation-invariant aggregator, and dual evaluators (element-level EE and global GE) whose scores are adaptively fused. The time complexity analysis and empirical experiments are also provided to assess the proposed PIETSP framework.

**Questions:**

Please refer to the weakness.

**Ethical Concerns:**

["NO or VERY MINOR ethics concerns only"]

**Final Justification:**

The paper needs significant improvement in its quality to reach the level of publication, including the paper writing and the comprehensiveness of the experiments. I will maintain my score to reflect the quality of paper.

**Limitations:**

Please refer to the weakness.

**Paper Formatting Concerns:**

I don't have any concerns.

**Quality:**

2

**Strengths And Weaknesses:**

Strengths
- Clear, mathematically grounded problem formulation. The paper recasts temporal-set prediction as a joint representation task that unifies element features and their time-varying membership.
- A time complexity analysis is provided to theoretically provide the backup for the efficiency claims, though the time complexity of baselines is not provided for reference or comparison.

Weaknesses:
- Figure 2 is difficult to interpret. For example, the “X” inside a circle is never defined. Important details are missing, so the figure is not self-contained and does not convey the workflow unambiguously.
- Lack of comparative complexity analysis. The paper presents only the time-complexity derivation for the proposed method and immediately claims efficiency. Without contrasting these results with state-of-the-art baselines, readers cannot assess how large—if at all—the computational advantage really is.
- Unclear source of the efficiency gains in Table 3. The baselines appear to be fully converged, yet the authors give no evidence that their own model was trained to convergence. Consequently, it is impossible to know whether the reported speed-ups come from fewer training epochs or from an intrinsically faster algorithm/architecture.
- Missing inference-time evaluation. A thorough comparison of inference latency and throughput against competing methods is strongly recommended to substantiate the paper’s claims of practical scalability.
- Clarity and organization need considerable refinement. The expression in the paper is often dense and poorly sign-posted, so grasping the main idea requires several close readings. Reordering sections for a more logical flow, defining symbols immediately where they appear, and adding concise transition sentences would greatly improve readability.

---

> ### Author Rebuttal · Authors · 2025-07-31
>
> We have underlined the core methods of earlier baselines which are RNN or GNN + Attention based which have quadratic complexity.
> Complexity analysis between latest baseline and our method is enumerated in Section 6.2 lines 284-289.
> In terms of training and convergence, we trained our model for a 100 epochs with an early stopping of patience=10 as enumerated in Section 4.7 lines 203-204. The model converged in much fewer epochs as recorded in Table 3.
> For inference time comparison, we have detailed the speed and throughputs of competing baselines in Table 3.

---

> > ### Comment · Reviewer_PNrR · 2025-08-07
> > **Official Comments from Reviewer PNrR**
> >
> > Thanks for the authors' reply. I don't see the reason to ignore the time complexity of other baselines discussed in this paper. Furthermore, the efficiency gain from the proposed method is still unclear. In Table 3, the gap in training epochs between the two methods is extremely large. The author did not reply to this question in detail. The paper is not well-written, such as Fig. 2. Thus, I will keep my score.

---

> ### Author Response · Authors · 2025-08-07
>
> We have clearly stated that complexity of baselines based on RNN or Attention+GNN are quadratic in number of unique set elements N, time-steps K or both. Compared to that our method is linear in both N and K.
>
> Efficiency gains are two fold: one based on the performance comparison on multiple datasets per Table 2. and second based on speed, throughput and epochs to converge based on Table 3.
>
> In Table 3, we did not set the training epochs for baselines, rather used their hyperparameter settings as is. The fact that state of the art baseline takes 300 epochs on an average to converge vs ours take around 10 itself is a huge efficiency gain.
>
> Kindly elaborate on which part is unclear regarding efficiency gain or time complexity and we can address those concerns.

---

> ### Comment · Reviewer_PNrR · 2025-08-08
> **Reply to Authors**
>
> Thank you for the authors' responses. However, some of the questions still remain unaddressed, and no responses are provided.
>
> - The paper presents time complexity analysis **only for one baseline**, as noted in Section 6.2. Perhaps my original question was unclear; I was eager to see a comparison of time complexity across ALL selected baselines included in the study. What is the reason the paper ignores the efficiency test among all other baselines in Table 3? Every paper should be self-contained and may have different settings. I understand that reusing other papers' results is somewhat in the gray area; however, at least the paper should state how this paper reuses or reproduces the results from other baselines. Unfortunately, the paper failed to provide even an efficiency assessment, including the latency test and time complexity analysis, among all baselines.
> - In addition, the authors repeatedly failed to provide explanations for Figure 2, yet I still find it unclear. This suggests that the writing and presentation of the paper require further improvement.
> - Thank you for the explanation of Table 3.

---

### Decision · Program_Chairs · 2025-09-17

**Decision:**

Reject

**Comment:**

This paper introduces PIETSP, a scalable permutation-aware model for temporal set prediction (TSP), using permutation-equivariant and permutation-invariant transformations combined with element-level and global evaluators. The method achieves linear time complexity in both sequence length and element count, and experiments on four benchmark datasets show performance comparable or superior to prior baselines, along with efficiency gains in training and inference.

Strengths. Reviewers agreed that the paper addresses a meaningful problem in dynamic set modeling with clear relevance to next-basket recommendation and temporal event forecasting (Reviewer pjY1, eDAC). The efficiency analysis and empirical speedups are convincing (Reviewer 16iN, eDAC). The work is well written in parts (Reviewer eDAC) and provides ablations of model components.

Weaknesses. However, all reviewers raised concerns about limited novelty, as the architecture builds directly on DeepSets-style permutation-equivariant/invariant blocks with relatively simple evaluators (Reviewer pjY1, 16iN, eDAC). The complexity analysis lacks breadth: authors present only their own derivation and compare efficiency mainly against SFCNTSP, omitting complexity contrasts with all baselines or recent sub-quadratic attention models (Reviewer PNrR, 16iN). Empirical evaluation, while broad in dataset coverage, is weakened by unclear sources of efficiency gains (Reviewer PNrR), missing error bars for baselines (Reviewer eDAC), and absence of comparisons with latest TSP methods such as LANET (Reviewer 16iN). Clarity issues remain around Figure 2 and notation (Reviewer PNrR, 16iN).

Discussion. Authors’ rebuttal clarified convergence protocols, added ablations, and argued that efficiency stems from modeling sequences holistically rather than set-by-set. Nonetheless, reviewers largely kept scores unchanged: Reviewer PNrR maintained borderline reject citing incomplete efficiency comparisons; Reviewer pjY1 appreciated clarifications but kept borderline reject due to weak novelty; Reviewer 16iN raised score slightly but still at borderline reject given missing comparisons; Reviewer eDAC held borderline accept but stressed lack of error reporting prevents substantiating SOTA claims. With an average rating of 3.25 and consensus leaning negative, the submission falls short of the NeurIPS acceptance bar.

Decision. While the paper tackles an important task and presents promising efficiency improvements, the contribution is undermined by limited methodological novelty, incomplete comparative analysis, and unresolved clarity concerns. The balance of strengths and weaknesses supports a Reject recommendation.

Follow-up for Authors. For future work, I encourage the authors to (1) include complexity and efficiency analysis across all relevant baselines, including recent sub-quadratic attention models; (2) provide error bars and statistical significance for both proposed and baseline methods; (3) compare against stronger, recent TSP architectures such as LANET and Set Transformers; (4) improve clarity of figures and notation; and (5) strengthen claims of novelty by positioning contributions beyond standard DeepSets formulations.

Final Recommendation: Reject.